# SARS-CoV-2 from COVID-19 Patients in the Republic of Moldova: Whole-Genome Sequencing Results

**DOI:** 10.3390/v14102310

**Published:** 2022-10-21

**Authors:** Alexandr Morozov, Vadim Nirca, Anna Victorova, Sven Poppert, Hagen Frickmann, Chiaki Yamada, Melissa A. Kacena, Sergiu Rata, Alexandru Movila

**Affiliations:** 1Molecular Biology Department, Imunotehnomed Ltd., MD-2001 Chisinau, Moldova; 2Laboratory of Systematics and Molecular Phylogeny, Institute of Zoology, MD-2028 Chisinau, Moldova; 3Diagnostic Department, Bernhard Nocht Institute for Tropical Medicine Hamburg, 20095 Hamburg, Germany; 4Department of Microbiology and Hospital Hygiene, Bundeswehr Hospital Hamburg, 20095 Hamburg, Germany; 5Institute for Medical Microbiology, Virology and Hygiene, University Medicine Rostock, 18057 Rostock, Germany; 6Department of Biomedical Sciences and Comprehensive Care, Indiana University School of Dentistry, Indianapolis, IN 46201, USA; 7Indiana Center for Musculoskeletal Health, Indiana University School of Medicine, Indianapolis, IN 46201, USA

**Keywords:** SARS-CoV-2, COVID-19, molecular epidemiology, Moldova, viral strains

## Abstract

Since the onset of the COVID-19 pandemic, no viral genome sequences of the severe acute respiratory syndrome coronavirus-2 (SARS-CoV-2) have been documented from the Republic of Moldova, a developing country geographically located in Eastern Europe between Romania and Ukraine. Here, we report the analysis of 96 SARS-CoV-2 sequences from Delta and Omicron variants of the SARS-CoV-2 cases in the Republic of Moldova obtained between August and November 2021 and between January and May 2022. Comparison to global viral sequences showed that among the Delta variant of the SARS-CoV-2, AY.122 (n = 25), followed by AY.4.2.3 (n = 6), AY.4 (n = 5), AY.43 (n = 3), AY.98.1 (n = 3), B.1.617.2 (n = 1), AY.125 (n = 1), AY.54 (n = 1), AY.9 (n = 1), AY.126 (n = 1), and AY.33 (n = 1) were the most frequently found lineages. Furthermore, 10 lineages of the Omicron variant, namely, BA.2 (n = 14), followed by BA.2.9 (n = 10), BA.1 (n = 5), BA.1.1 (n = 5), BA.1.18 (n = 4), BA.1.15.1 (n = 3), BA.1.17.2 (n = 2), BA.1.17 (n = 2), BA.1.15 (n = 1), and BA.2.1 (n = 1) were detected. In addition, we also identified the impact of the military crisis between Russia and Ukraine, when the COVID-19 epidemiological rules collapsed, on the distribution of Delta and Omicron variants in the Republic of Moldova. Additional studies are warranted to characterize further the impact of the war between Russia and Ukraine on the genomic epidemiology of the SARS-CoV-2 in the Republic of Moldova and Eastern Europe.

## 1. Introduction

For more than two years, the spread of the pandemic of the severe acute respiratory syndrome coronavirus-2 (SARS-CoV-2), the causative agent of coronavirus disease 2019 (COVID-19), remains a relevant threat affecting health policy and social life worldwide. The first precedents of a new type of pneumonia of unknown etiology, later causally attributed to SARS-CoV-2, were reported in December 2019 in Wuhan, China [1]. On 7 January 2020, scientists identified the SARS-like beta-corona-(Sarbeco-) virus [2]. The discovery of the pathogen and its molecular genetics formed the scientific basis for the development of various diagnostic test assays, beginning with early available in-house PCR assays, which were either published in the international literature or by public health organizations [3]. Beyond diagnostic needs, phylogenomic analyses are essential for understanding and forecasting the risks of the COVID-19 pandemic.

Because of the global spread of the SARS-CoV-2, the Republic of Moldova was also significantly impacted by the COVID-19 pandemic. The first cases of SARS-CoV-2 infection in the Republic of Moldova were confirmed on 7 March 2020 [4]. Until June 2022, the total number of recorded COVID-19 cases in Moldova was ≈520,000. According to the Moldavian Ministry of Health, from the beginning of the pandemic until June 2022, 11,555 death cases were reported. Of note, more than a million people (≈40% of the population) had been fully vaccinated, and about 83,000 people were also re-vaccinated [5].

In June 2022, 13 genomic variants comprising seven mainly significant variants of the new coronavirus were circulating unevenly distributed worldwide [6]. Since the publication of the first whole genome of SARS-CoV-2 in January 2020, numerous genome sequences have been obtained and shared by researchers worldwide. In June 2022, more than 11,421,000 cases of SARS-CoV-2 genome sequences were received in more than 200 countries and deposited in public databases such as GISAID (Global Initiative on Sharing Avian Influenza Data). Consequently, the global spread and phylodynamics of SARS-CoV-2 are under surveillance, and various lineages have been designated and determined. Prior to the COVID-19 pandemic, the high costs associated with next-generation sequencing assay and the small population of the Republic of Moldova made it unfeasible to create a genetic laboratory equipped with whole genome sequencing equipment. Therefore, no whole genome SARS-CoV-2 sequences from Moldova have been analyzed in detail.

During a selected assessment period between August 2021 and November 2021, Moldova experienced a notable increase in the number of people infected with SARS-CoV-2 and of patients hospitalized due to COVID-19 [7]. This was the third wave of COVID-19 in Moldova. During this time, global circulation of the delta variant was observed, associated with considerable risks to individuals and health care systems. The delta variant has been classified in several nomenclature systems. According to the Pango nomenclature, it became known as B.1.617.2. In addition to this classification, this variant has several lineages named AY.x (the “x” is a number that varies from 1 to 131, according to the lineage). 

The Omicron variant (also known as the progeny of B.1.1.529) of the SARS-CoV-2 was discovered in the Republic of Botswana in early November of 2021. Then, South Africa reported it to the World Health Organization on 24 November 2021, and it was designated as a variant of concern (VOC) on 26 November 2021 [8]. The Omicron variant contains a large number of previously documented mutations in other VOCs, including at least 32 mutations in the spike protein alone, compared to the 16 mutations in the already highly infectious Delta variant of the SARS-CoV-2. By March 2022, the Omicron variant of SARS-CoV-2 became the only actively circulating variant of concern in the Republic of Moldova [9].

For the first time in the Republic of Moldova, this study characterized the genomic information, determined mutations, and conducted phylogenomic analyses of 96 SARS-CoV-2 genomes extracted from samples collected between August 2021 and May 2022 at the MedExpert private laboratory center, Chisinau, Republic of Moldova. In addition, we also evaluated the impact of the Ukrainian military crisis induced by Russia, when the Ukrainian refugees entered Moldova, on the SARS-CoV-2 distribution in Moldova. 

## 2. Materials and Methods

### 2.1. Sample Collection and Selection

Samples were chosen from the archive of the private medical laboratory MedExpert (Chisinau, Republic of Moldova), one of the five laboratories certified to perform SARS-CoV-2 testing in Moldova. During the entire period of the pandemic, starting from March 2020, all positive human nasopharyngeal and oropharyngeal swabs from all regions of Moldova were stored at −70 °C. We chose two periods: August–November 2021 and January–May 2022. The first period was selected as the period preceding the refugee crisis, including restrictions on self-isolation that were no longer so strict, wherein regular flights were resumed. The second period was chosen in the window of time when the change of Delta to Omicron took place and when the refugee crisis from Ukraine occurred. We analyzed all available sequences, but for a detailed phylogenetic analysis, we selected 48 samples from each period and analyzed them. In the discussion part, we consider all the sequences available on GisAid for the selected periods for all countries of interest in the context of Moldova.

### 2.2. Extraction of Total Nucleic Acids and qRT-PCR

The study samples were randomly selected on the basis of the cycle threshold-(Ct-) values obtained by real-time PCR during the initial diagnostic screening. Inclusion criteria comprised a Ct-value range from 15 to 28. The deep-frozen samples were thawed, and nucleic acids were extracted using the PREP-NA-S commercial kit according to the manufacturer’s instructions (DNA technology, Russian Federation, cat. N° P-007-N/1INT). Samples were tested again using a commercial RUO (research use only) real-time PCR kit from BioPerfectus technologies (China, cat. N° JC10315NW-50T). Amplification was performed on a DtPrime Real-Time PCR device (DNA technology, Moscow, Russian Federation). As detailed below, samples with a reproducible Ct-value were selected for further Nanopore sequence analyses.

### 2.3. cDNA Synthesis and Multiplex PCR

cDNA synthesis was performed using the LunaScript^®^ RT SuperMix Kit (New England Biolabs, Ipswich, MA, USA) according to the manufacturer’s instructions. The cDNA was amplified using the multiplex MIDNIGHT protocol scheme for SARS-CoV-2 (v.4), composed of two primer pools with 58 primers. The following reaction mix was used for the multiplex PCR:

2.5 μL template cDNA;3.67 μL nuclease-free water;6.25 μL 5× Q5 HS Master Mix (New England Biolabs, Ipswich, MA, USA)0.63 μL primer pool 1 or 2.

The cycling conditions comprised an initial activation step at 98 °C for 30 s, followed by 35 cycles of denaturation at 98 °C for 15 s and of annealing and amplification at 65 °C for 5 min, as well as a final cooling down to 4 °C. The amplified pools 1 and 2 were subsequently combined, and rapid barcodes (SQK-RBK 110.96, Oxford Nanopore Technologies, London, UK) were added. Finally, the amplified, mixed, and barcoded pools 1 and 2 were purified with magnetic beads (Nimagen, Nijmegen, The Netherlands) at a ratio of 1:1.

### 2.4. Nanopore Library Preparation

The preparation of each library for the Oxford Nanopore platform was performed according to the MIDNIGHT protocol by applying ligation reagents from the provided kit (SQK-RBK 110.96, Oxford Nanopore Technologies). After preparation, the library was loaded on the R 9.4.1 flow cell (Oxford Nanopore Technologies, London, UK) and sequenced on a MinION Mk1B device (version 21.10.4) (Oxford Nanopore Technologies) for 18 h.

### 2.5. Reference-Based Assembly

The obtained raw data from the MinION Mk1B device were base-called with the “guppy_basecaller” software and demultiplexed with the guppy_barcoder software (both part of “Guppy base calling suite (C)”, Oxford Nanopore Technologies, Limited. Version 6.0.1+652ffd179). The demultiplexed sequences were then analyzed using the EPI2ME wf-artic pipeline running on the Nextflow software platform (version 21.10.4.5656) [10].

### 2.6. Data Deposition

The genome sequences and associated metadata are published in the GISAID’s EpiCoV database. To view each individual sequence with details such as accession number, virus name, collection date, originating lab, and submitting lab, visit https://doi.org/10.55876/gis8.220615cz (accessed on 1 August 2022).

Data Snapshot: EPI_SET_20220615cz is composed of 96 individual genome sequences. The collection dates range from 2021-08-09 to 2022-05-10. Data were collected in the Republic of Moldova. All sequences in this dataset are compared to the hCoV-19/Wuhan/WIV04/2019 (WIV04), the official reference sequence employed by GISAID (EPI_ISL_402124).

### 2.7. Phylogenomic Analysis

Altogether, 96 samples were included in the phylogenomic analysis. The phylogenomic tree alignment was performed with the MEGA 11 software [11] using the Tamura–Nei maximum likelihood (ML) method with a 1000-fold bootstrap [12,13]. The Nextclade Pangolin lineage option was used when determining lineages using the Nextlade software, which was identified by applying a Nextclade web tool (https://clades.nextstrain.org (accessed on 1 August 2022)) [6]. 

### 2.8. Ethical Clearance

The study was conducted under the Moldavian Code of Ethics of the medical worker and pharmacist approved by the Government Decision no. 192 of 24.03.2019, LAW no. 411 of 28.03.1995 on healthcare, LAW no. 264 of 27.10.2005 regarding the exercise of the medical profession, LAW no. 263 of 27.10.2005 regarding the rights and responsibilities of the patient, and LAW on the protection of personal data no. 133 of 08.07.2011. Furthermore, the data included in this study were openly available to the public before the initiation of the study.

## 3. Results

Altogether, 96 whole-genome sequences were obtained from SARS-CoV-2-positive samples collected from humans. Forty-eight samples were collected between August 2021 and November 2021, and an additional 48 samples were collected between February 2022 and May 2022 in Moldova. The sequences were ≈29,800 nucleotides in length, with a mean read depth ranging from 80 to 200. Samples collected in 2021 were all identified as the Delta variant; samples collected in 2022 were identified as the Omicron variant of SARS-CoV-2. Detailed information about available genome sequences is provided in Appendix A.

### 3.1. The Delta Variant of SARS-CoV-2

Altogether, 291 distinct single-nucleotide substitutions (SNSs) (Appendix A) were recorded by aligning with the first Wuhan complete sequence in the GISAID database (EPI_ISL_402124). All parsed sequences collected from August 2021 to November 2021 were identified as Delta variants of SARS-CoV-2 (Appendix A). Ten lineages and sub-lineages of the variant were observed (Figure 1a). Among those, the most frequently found lineages were AY.122 (n = 25), followed by AY.4.2.3 (n = 6), AY.4 (n = 5), AY.43 (n = 3), AY.98.1 (n = 3), B.1.617.2 (n = 1), AY.125 (n = 1), AY.54 (n = 1), AY.9 (n = 1), AY.126 (n = 1), and AY.33 (n = 1). Almost all assessed samples belonged to the clade 21J (Delta) (n = 46). However, there were two exceptions. The sample EPI_ISL_7729028 belonged to the clade 21I (Delta) lineage AY.9, and the sample EPI_ISL_7778879 to the clade 21A (Delta) lineage AY.54.

The majority of the sequenced Moldovan Delta-positive samples showed the following combination of mutations: Spike T478K, Spike T19R, R158del, Spike P681R, Spike G1D, Spike L452R, Spike F157del, NSP4 V167L, NSP3 P1469S, NSP3 A488S, NSP14 A394V, NSP13 P77L, NSP12 P323L, NSP12 G671S, NS7b T40I, NS7a V82A, NS7a T120I, NS3 S26L, N R203M, N G215C, N D63G, N D377Y, and M I82T.

The 291 SNS (Appendix A) included 68% transitions and 22% transversions. Transitions were distributed as follows: C > T (42.96%), T > C (10.31%), A > G (8.93%), and G > A (5.84%), whereas transversions were as follows: G > C (24.05%), followed by other transversions with proportions of less than 1.5% each (Appendix A). The most frequent substitution was the transversion C > T (42.96%), and the least frequently observed one was G > C (0.69%).

Among this 291 SNSs, 177 led to amino acid exchanges (aaSNS), of which 51 aaSNS occurred in the ORF1a sequence, 38 aaSNS in the ORF1b sequence, 31 were found in the S gene, 19 in the ORF3a sequence, 13 in the N gene, 8 in the ORF7a sequence, 8 in the ORF8 sequence, 4 in the ORF9b sequence, 2 in the M gene, 2 in the ORF6 sequence, and 1 in the ORF7b sequence.

### 3.2. The Omicron Variant of SARS-CoV-2

Almost 50% of the 48 samples belonged to clade 21K (Omicron) (n = 22), and 25 samples belonged to clade 21L (Omicron) (Appendix A). One examined sample was identified as the recombinant type (hCoV-19/Moldova/G0006/2022).

Ten lineages of the variant were observed (Figure 1b). Among those, the most frequently found lineages were BA.2 (n = 14), followed by BA.2.9 (n = 10), BA.1 (n = 5), BA.1.1 (n = 5), BA.1.18 (n = 4), BA.1.15.1 (n = 3), BA.1.17.2 (n = 2), BA.1.17 (n = 2), BA.1.15 (n = 1), and BA.2.1 (n = 1). 

The vast majority of the sequenced Moldovan Omicron-positive samples showed the following combination of mutations: Spike A27S, Spike D405N, Spike D614G, Spike D796Y, Spike E484A, Spike G142D, Spike G339D, Spike H655Y, Spike K417N, Spike L24del, Spike N440K, Spike N501Y, Spike N679K, Spike N764K, Spike N969K, Spike P25del, Spike P26del, Spike P681H, Spike Q493R, Spike Q498R, Spike Q954H, Spike R408S, Spike S373P, Spike S375F, Spike S477N, Spike T19I, Spike T376A, Spike T478K, Spike V213G, Spike Y505H, E T9I, M A63T, M Q19E, N G204R, N R203K, N S413R, NS3 H78Y, NS3 T223I, NSP1 S135R, NSP3 G489S, NSP3 T24I, NSP4 L264F, NSP4 T327I, NSP4 T492I, NSP5 P132H, NSP6 F108del, NSP6 G107del, NSP6 S106del, NSP12 P323L, NSP12 S451N, NSP13 R392C, NSP14 I42V, and NSP15 T112I.

A total of 216 single-nucleotide substitutions (SNS) (Appendix A) were recorded, including 77% transitions and 23% transversions. Transitions were distributed as follows: C > T (42.59%), T > C (8.8%), A > G (14.81%), and G > A (11.57%), whereas transversions were as follows: G > C (2.87%), followed by other transversions with proportions of less than 5% each (Appendix A). The most frequent substitution was the transversion C > T (42.59%), and the least frequently observed one was C > G (0.69%).

Among the 216 SNSs (Appendix A), 117 led to amino acid exchanges (aaSNS), of which 40 aaSNS occurred in the ORF1a sequence, 35 were found in the S gene, 19 in the N gene, 15 aaSNS in the ORF1b sequence, 9 in the ORF3a sequence, 4 in the M gene, 3 in the ORF9b sequence, 3 in the ORF6 sequence, 2 in the E gene, and 1 in the ORF7a sequence.

## 4. Discussion

The Republic of Moldova has certain features that distinguish it from other countries. According to the Migration Policy Institute (MPI), the population of Moldova is estimated at 3.2 million people, of which 1.2 million people work outside of Moldova, which is about a third of the country’s total population [14]. Of these 1.2 million emigrants, 935 thousand (935 k) people (≈80%) come from four countries: Russia (≈300 k), Romania (≈285 k), Italy (≈200 k), and Ukraine (≈150 k) [14]. Emigrants working in these countries regularly visit Moldova to visit relatives and friends. Moldova is also the least visited non-microstate country in Europe [15]. Taking into account the statistics of border crossings [16], we expected to see coronavirus lineages distribution somewhere between these four countries. The combination of these factors and the geographical position of Moldova creates special epidemiological conditions.

The Delta variant became a variant of interest on 4 April 2021, and a variant of concern on 11 May 2021 [17]. The variant had more than 14 mutations; the most relevant ones compromised the spike protein of the virus [18].

Three clades are distinguished within the Delta variant. In the study presented here, all three Delta clades were found, comprising 21J (Delta) with eight lineages, 21I (Delta) lineage AY.9, and clade 21A (Delta) lineage AY.54 for the reference period between August and November 2021 (Figure 2). In Russia, AY.122 was dominant, occupying 87% of the total number of studied sequences. In Romania and Italy, it was observed that AY.122 occupies only 9% and 11% of cases, respectively. In contrast, the AY.43 lineage was dominant over the selected period, occupying 28% and 24%, followed by AY. 4 with 25% and 10% in Romania and Italy, respectively. Interestingly, the AY.4.2.3 sub-lineage was present in Moldova (12%), while in the other countries, this line consisted of an insignificant part (≈1%). However, a detailed examination of sequences from Romania for the selected period (August–November 2021) showed AY.4.2.3 takes 0.99% (n = 51) (EPI_SET_220916n). Still, more than half of these sequences were obtained from patients from the Iasi and Vaslui counties that border Moldova. 

To our knowledge, the SARS-CoV-2 genetic distribution in Ukraine demonstrated high similarity to those virus strains observed in Moldova. Ukraine, like Moldova, also has close ties with Russia in the east and with Europe in the west. The proportion rates for lineages AY.122 and AY.4 for Ukraine and Moldova are similar—59% and 5% for Moldova and 57% and 4% for Ukraine, respectively. Furthermore, the identified nucleotide substitutions included all major hotspot mutations, which are characteristic of the Delta variant of the virus [19]. The most significant of these mutations were P681R—a mutation that can also be associated with a decrease in neutralizing antibodies [20], and Pro323Leu (n = 44, 92% in the present study) with notably higher infectivity and transmissibility rates than those recorded for the older variants [21]. Asp614Gly (n = 48/48) in the Spike-protein-associated SNS often co-occurs with three others [22] that were also present in the assessed Moldovan samples. These SNSs are C241>T (n = 48; 100%), C3037>T (n = 48; 100%), and C14408>T (n = 44; 92%). In four samples, the substitution in the case of the SNS C14408> was not >T, but >G, which led to a change to arginine instead of leucine.

The second investigated period (January–May 2022, Appendix A) differs from the first due to the tragic events associated with the military conflict between Ukraine and Russia in February 2022. The invasion caused the largest refugee crisis in Europe since World War II when over 606,000 refugees crossed the Moldova–Ukrainian border in a short period [23]. In addition, the air traffic between Russia was immediately terminated, which also significantly reduced the passenger turnover. Therefore, the Moldavian genetic variants were BA1.1 (29%), BA.2.9 (11%), and BA.2 (8%) while BA.2.9 (53%) and BA.1.1 (17%) were dominant in Ukraine prior to the refugee crisis in February 2022. By contrast, the Ukrainian BA.2.9. became dominant in Moldova in March 2022 (Figure 3).

BA.2.9 differs from BA.2, with only one AA change (ORF3a: H78Y) and two nucleotide changes (C22792T, C25624T). There is no evidence that BA.2.9 has an increase in transmissibility, in infectious duration, or in immune evasion. Thus, we assume that both BA.2.9 and BA.2 have an equal rate of relative growth. In Moldova and Poland, the initial proportion of the BA.2 variant in February (8% and 13%, respectively) was higher than BA.2.9 (6% and 11%, respectively). However, in March, BA.2.9. became dominant (48% and 41%, respectively), and therefore the reasonable explanation is that the initial proportion significantly changed, which could have occurred because of the refugee crisis.

When matching with the Delta variant, the Omicron variant exhibits two main and extrinsically interesting molecular features: a consolidation of deletions and insertion and an enduringly increased mutational landscape in the spike region. Conserved AA substitutions (G142D, P681L, D614G) demonstrate an interesting evolutionary penetration in both variants. For example, D614G appears to be neuro-invasive, entering the central nervous system via the olfactory nerve, as previously observed in other studies [24,25]. The observed neurovirulence associated with D614G infection included antiviral and inflammatory responses in the olfactory bulb and, to a lesser extent, in the cerebral cortex [26].

Within the Omicron variant, we observed the simultaneous presence of D614G, but the lack of E484K was observed in Delta samples. The mutation persists in the same locus, but the AAs are changed to E484Q. That combination (D614G/ E484Q) seems to be unique, leading the virus to significantly high infectivity/transmissibility and potentially to decreased response rates to mAbs and targeted vaccines [27,28].

As with all studies, this study has several limitations. First, only a restricted number of samples were able to be analyzed during a defined and short period. Second, a potential bias could have resulted from the fact that only samples with high copy numbers could be included for technical reasons because it is hypothetically possible that certain genotypes are more likely to be associated with low Ct values, as observed in the included samples.

## 5. Conclusions

This study provides the first epidemiological overview of the abundance of Delta and Omicron genetic lineages of the SARS-CoV-2 virus in the Republic of Moldova associated with the military crisis occurring in Ukraine when the COVID-19 epidemiological rules in Europe collapsed. 

## Figures and Tables

**Figure 1 viruses-14-02310-f001:**
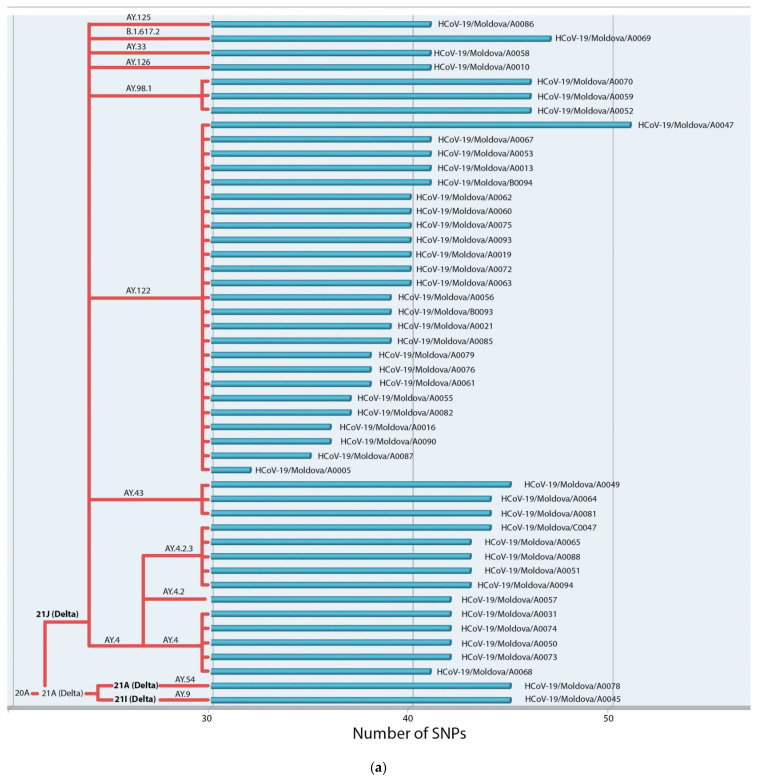
(**a**) Number of the SNPs and classification of the Delta variant of SARS-CoV-2 isolates from Moldova according to the Nextclade Pango Lineage nomenclature compared to the Wuhan complete genome sequence. (**b**) Number of SNPs and classification of the Omicron variant of SARS-CoV-2 isolates from Moldova according to the Nextclade Pango Lineage nomenclature compared to the Wuhan complete genome sequence.

**Figure 2 viruses-14-02310-f002:**
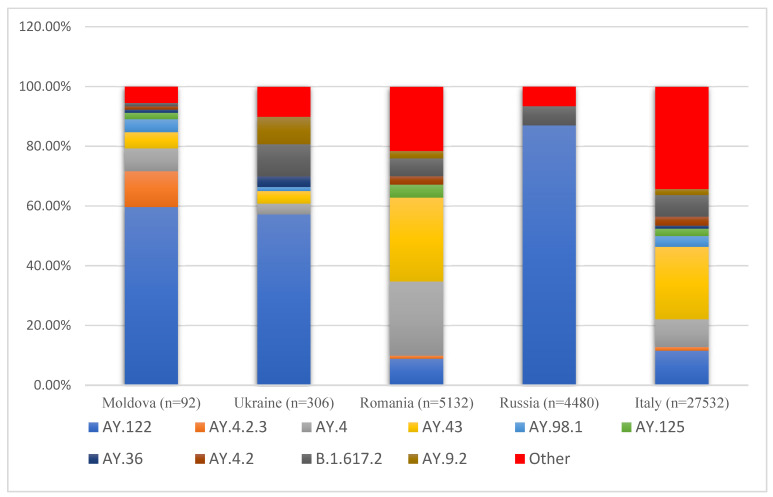
Distributions of SARS-CoV-2 lineages in the period August–November 2021.

**Figure 3 viruses-14-02310-f003:**
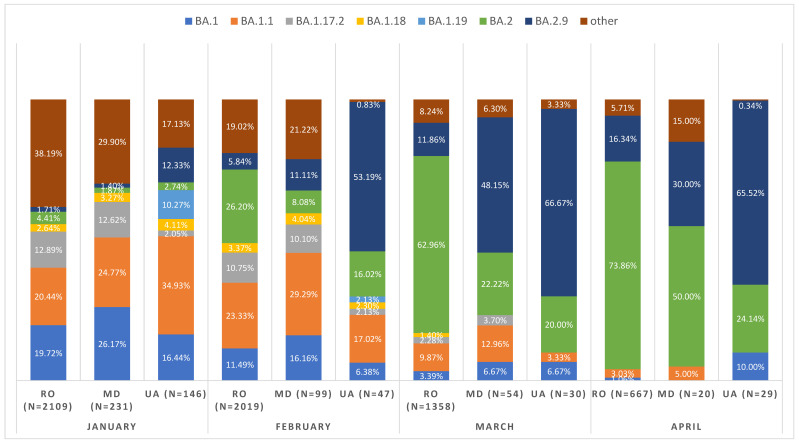
Monthly distribution of SARS-CoV-2 lineages in January-April 2022. (RO—Romania, MD—Moldova, UA—Ukraine).

## Data Availability

The sequences of SARS-CoV-2 that are reported in this study are deposited in the database of the Global Initiative on Sharing Avian Influenza Data (GISAID), available at https://www.gisaid.org (accessed on 15 August 2022) in EpiCoV database. Otherwise, all relevant information is provided in the article. Raw data can be made available at reasonable request.

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
