# Peer review of "SARS-CoV-2 from COVID-19 Patients in the Republic of Moldova: Whole-Genome Sequencing Results"

_viruses, 2022, doi:10.3390/v14102310_

Round 1

Reviewer 1 Report

The current study analyzed 96 genomes collected from August 2021 - May 2022, sequenced by Nanopore technology and analyzed using Pango lineages and phylogenetic analysis, describing the variants in Moldova at the time.

Major comments:
1. The main problem with the way the manuscript is presented, is that it announces that 96 samples from Moldova have been sequenced, some from a long time ago, and describes the results in terms of variants and detailed mutations. It seems like the manuscript is missing the implications of the findings, indication of how or why these data are relevant to Moldova or even the world. E.g - how do these findings compare to other countries around Moldova at the time, how is it affected by importation of sars-cov-2 into the country at the relevant times, implications to clinical situation of the patients in the country, and more..
2. A summary of the variant findings on a longitudinal graph and / or a even a piechart or even a table showing what variants were found and in what times is highly missing. Its very hard to understand all this information from the lists in the text, or understand the relevance.
3. The authors constructed nice phylogenetic trees, but do not discuss them. Do they show evidence of transmission, what do they imply, other than the clade (which can be also given in Nextclade, e.g.). A minor comment in this respect, is that the nomenclature used for the clades (21J e.g.) is nextstrain nomenclature, while the nomenclature used for the variants (B.1.617.2 e.g.) is pangolin nomenclature, which not always go hand in hand, and this was not explained. Also, it is more precise nowadays to determine the correct variant using Usher mode for pangolin, which is not clear if that was done here.

Minor comments:
1. The manuscript could use an extensive editing of the english language.
2. The authors do not mention how the 96 samples were chosen and why, out of the many available samples for sequencing.

Author Response

We are greatly indebted to Reviewer 1’s comments. Following those constructive comments, we revised the article as responsibly as possible. 

Major comments:

  1. The main problem with the way the manuscript is presented, is that it announces that 96 samples from Moldova have been sequenced, some from a long time ago, and describes the results in terms of variants and detailed mutations. It seems like the manuscript is missing the implications of the findings, an indication of how or why these data are relevant to Moldova or even the world. E.g - how do these findings compare to other countries around Moldova at the time, how is it affected by importation of sars-cov-2 into the country at the relevant times, implications to clinical situation of the patients in the country, and more..

Answer: We have significantly changed the discussion and added the requited compared to other countries around Moldova. In addition, we have also examined the potential impact of the Ukrainian refugee crisis on the SARS-CoV-2 distribution in Moldova.

  1. A summary of the variant findings on a longitudinal graph and / or an even a piechart or even a table showing what variants were found and in what times is highly missing. It’s very hard to understand all this information from the lists in the text, or understand the relevance.

Answer: Missing supplementary tables were provided (Supplementary tables 1-4.)

  1. The authors constructed nice phylogenetic trees, but do not discuss them. Do they show evidence of transmission, what do they imply, other than the clade (which can be also given in Nextclade, e.g.). A minor comment in this respect, is that the nomenclature used for the clades (21J e.g.) is nextstrain nomenclature, while the nomenclature used for the variants (B.1.617.2 e.g.) is pangolin nomenclature, which not always go hand in hand, and this was not explained. Also, it is more precise nowadays to determine the correct variant using Usher mode for pangolin, which is not clear if that was done here.

Answer: We conducted the analysis as suggested by the Reviewer 1 using the software Nextlade, which were identified by applying a Nextclade web tool (https://clades.nextstrain.org). We used the Nextclade Pangolin lineage option when performing the phylogenomic analysis. «2.7. Phylogenomic analysis» part in m&m was adjusted.
The usher mode for not used in current study.

As it is mentioned in the following article https://docs.nextstrain.org/projects/nextclade/en/stable/user/algorithm/nextclade-pango.html

The nextclade pango

Minor comments:

  1. The manuscript could use an extensive editing of the english language.

Answer: We have used Grammarly to edit English.

2, The authors do not mention how the 96 samples were chosen and why, out of the many available samples for sequencing.

Answer: The detailed explanation was provided as suggested by Reviewer 1. Please see the section 2.1. Sample collection and selection”

Reviewer 2 Report

The paper by Morozov et al. presents the results of whole genome sequencing of only 96 sequences to represent one country over 10 months. The size of these data seems to me to be at odds with the size given and written by the authors in the abstract.

The sentence line 28 of the abstract raises questions the pathogenicity of Spike at position D614G has already been reported and long before the advent of the Delta variant. it appears to be unrelated to the substance of the paper and the authors' main findings.

Introduction section:

-the first paragraph can be rewritten by the authors to be shorter and can go more to the point.

3 Results section:

96 WGS with no gaps in sequencing ? all NNN ? all Frameshift ? authors can give more details on their high quality genomes.

Materials and methods

Maximum likelihood is more consistent with phylogenomic analyses than neighbour-joining.

SNS is confusing: do authors detect nucleotide mutations or amino acid substitutions? Please clarify this point, as well as in other parts of the manuscript.

Line 190 Spike is mentioned twice for the T19R mutation.

Line 191 Spike G1D is mentioned twice and furthermore this sentence is wrong, spike like other proteins starts with a methionin aminoacid M .

The abstratc needs to be rewritten and more adapted to the result especially on Omicron.

The phylogenomic tree should have scales.

Figure legends

Phylogenomic trees instead of phylogenetic trees.

The discussion section is too long. 

The authors cannot discuss all the mutations that are described in detail in other papers. They have to focus on the points that seem essential to them in the diversity observed in Moldova, probably as they did for the 323 substitution.

There is no data on the dynamics of these infections and clusters. There is little exploitation of the data except in the phylogenomic tree. Overall the manuscript requires careful reading.

Author Response

We are greatly indebted to Reviewer 2’s comments. Following those constructive comments, we revised the article as responsibly as possible. 

1.-the first paragraph can be rewritten by the authors to be shorter and can go more to the point.

Answer. The first paragraph was changed as suggested.

  1. 96 WGS with no gaps in sequencing ? all NNN ? all Frameshift ? authors can give more details on their high quality genomes.

Answer: The detailed table was provided to the Supplementary part (Supplementary Table S3.)

3.Maximum likelihood is more consistent with phylogenomic analyses than neighbour-joining.

Answer: We did evolutionary analysis by Maximum Likelihood method, new phylogenetic trees were provided in the manuscript.

4.SNS is confusing: do authors detect nucleotide mutations or amino acid substitutions? Please clarify this point, as well as in other parts of the manuscript. 

Answer: As it is showed in Tables S1 and S2 we have collected data and show the data on both nucleotide mutations and aa substitutions.

5.Line 190 Spike is mentioned twice for the T19R mutation.

Answer: Corrected

  1. Line 191 Spike G1D is mentioned twice and furthermore this sentence is wrong, spike like other proteins starts with a methionin aminoacid M .

Answer: Corrected

7.The abstract needs to be rewritten and more adapted to the result especially on Omicron.

Answer: Corrected

  1. The phylogenomic tree should have scales.

Answer: We did evolutionary analysis by Maximum Likelihood method, new tree with scales was added to the manuscript.

  1. Phylogenomic trees instead of phylogenetic trees.

Answer: Corrected

  1. The discussion section is too long.  The authors cannot discuss all the mutations that are described in detail in other papers. They have to focus on the points that seem essential to them in the diversity observed in Moldova, probably as they did for the 323 substitutions.

Answer: We have significantly modified the discussion part as was suggested by the Reviewer.

  1. There is no data on the dynamics of these infections and clusters. There is little exploitation of the data except in the phylogenomic tree. Overall, the manuscript requires careful reading.

Answer. The requested data and discussion are provided.

Reviewer 3 Report

In this study, 96 complete SARS-CoV-2 genomes were characterized by genomic sequencing, where mutations were determined and phylogenetic analyzes were performed on these strains extracted from samples collected between August 2021 and May 2022 at the MedExpert private laboratory center, Chisinau, Republic of Moldova. All samples collected in 2021 were identified as Delta variant; samples collected in 2022 were identified as the Omicron variant of SARS-CoV-2.

Regarding the Delta variant, the authors found 291 distinct single nucleotide substitutions (SNSs) and 216 SNSs in the Omicron variant when compared to the first complete sequence from Wuhan, pinpointing combinations of mutations throughout the SARS-CoV-2 genome. However, it is unclear whether the authors found anything significant in the strains isolated from their populations.

Author Response

We are greatly indebted to Reviewer 3 comments. As was suggested, we have compared our results to Moldova’s neighbor countries to show the significant changes in lineage distribution.

Reviewer 4 Report

The manuscript is modified as per the suggestions, it can be accepted after minor revision.

Figure 2 is not visible clearly and figure captions should be written properly.

Author Response

We are greatly indebted to Reviewer 4 comments. We have corrected all figures as suggested by the Reviewer.

Round 2

Reviewer 1 Report

The authors have modified the manuscript according to the major comments suggested. Still it seems like the figures and tables the authors chose to display in the manuscript vs. the supplementary are not appropriate. Eg., Table 1 which lists all the SNS found - the meaning of this is not discussed or clear.. what do all these mutations mean? did they find new mutations in Moldova vs. other places? which ones belong to the variant lineage and which ones do not?  it's better to put one of the graphs from the supplementary instead, e.g. Figures S1-4, which describe the variants in Moldova and are very clear. Same with the lineage trees - they belong in the supplementary since they do not provide any additional information and are not really discussed anywhere. 

Author Response

As the reviewer suggested, we have moved Table 1 to supplementary material labeling it as Supplementary Table 4. The main goal of the table is to demonstrate the transversions difference between Delta and Omicron variants (G > T, for example).
In addition, Figures 2A and 2B were also moved to the supplementary files as suggested (Now it is Supplementary Figures S1 A &B). However, we still want to leave Figure 1 in the main text because this data is essential to demonstrate the number of mutations found in SARS-CoV-2 variants.

In addition, we relocated Supplementary Figure S1 & Supplementary Figure S3 to the main text labeling them as Figure 1 and Figure 2.

Reviewer 3 Report

Line 225, it is important that you review the sample number corresponding to AY.125 lineage 

Author Response

Thank you for identifying this error; we have fixed it in the text. The corrected number for AY.125 is n=1.